# Identification of the Most Suitable Mobile Apps to Support Dietary Approaches to Stop Hypertension (DASH) Diet Self-Management: Systematic Search of App Stores and Content Analysis

**DOI:** 10.3390/nu15153476

**Published:** 2023-08-06

**Authors:** Ghadah Alnooh, Tourkiah Alessa, Essra Noorwali, Salwa Albar, Elizabeth Williams, Luc P. de Witte, Mark S. Hawley

**Affiliations:** 1Centre for Assistive Technology and Connected Healthcare, School of Health and Related Research, University of Sheffield, Sheffield S1 2NU, UK; gsaalnooh1@sheffield.ac.uk; 2Department of Health Sciences, College of Health and Rehabilitation Sciences, Princess Nourah Bint Abdulrahman University, Riyadh 11671, Saudi Arabia; 3Department of Biomedical Technology, College of Applied Medical Science, King Saud University, Riyadh 11433, Saudi Arabia; talessa@ksu.edu.sa; 4Clinical Nutrition Department, Faculty of Applied Medical Sciences, Umm Al-Qura University, Makkah 21955, Saudi Arabia; eanoorwali@uqu.edu.sa; 5Food and Nutrition Department, Faculty of Human Sciences and Design, King Abdulaziz University, Jeddah 21589, Saudi Arabia; salbar1@kau.edu.sa; 6Department of Oncology and Metabolism, The Medical School, University of Sheffield, Sheffield S1 2NU, UK; e.a.williams@sheffield.ac.uk; 7Research Group Technology for Healthcare, Centre of Expertise Health Innovation, The Hague University of Applied Science, 2521 EN Den Haag, The Netherlands; l.p.dewitte@hhs.nl

**Keywords:** DASH diet, hypertension, self-management, smartphone app, mHealth, behaviour change techniques, quality

## Abstract

Smartphone apps might provide an opportunity to support the Dietary Approaches to Stop Hypertension (DASH) diet, a healthy diet designed to help lower blood pressure. This study evaluated DASH diet self-management apps based on their quality, likely effectiveness, and data privacy/security to identify the most suitable app(s). A systematic search and content analysis were conducted of all DASH diet apps available in Google Play and the Apple App Store in the UK in November 2022. Apps were included if they provided DASH diet tracking. A previous systematic literature review found some commercial apps not found in the app store search, and these were also included in this review. Three reviewers used the App Quality Evaluation Tool (AQEL) to assess each app’s quality across seven domains: knowledge acquisition, skill development, behaviour change, purpose, functionality, and appropriateness for adults with hypertension. Domains with a score of 8 or higher were considered high-quality. Two reviewers assessed the apps’ data privacy and security and then coded Behaviour change techniques (BCTs) linked to the Theoretical Domain Framework (TDF) underpinning the likely effectiveness of the apps. Seven DASH diet apps were assessed, showing the limited availability of apps supporting DASH diet self-management. The AQEL assessment showed that three apps scored higher than eight in most of the AQEL domains. Nineteen BCTs were used across the apps, linked to nine TDF action mechanisms that may support DASH diet self-management behaviours. Four apps met standards for privacy and security. All seven apps with self-monitoring functionality had sufficient theoretical basis to demonstrate likely effectiveness. However, most had significant quality and data security shortcomings. Only two apps, NOOM and DASH To TEN, were found to have both adequate quality and security and were thus deemed suitable to support DASH diet self-management.

## 1. Introduction

Hypertension is one of the leading causes of increased morbidity and mortality worldwide [1] due to the negative impact of high blood pressure (BP) on cardiovascular and renal functions [2,3]. Elevated BP levels are caused by a complex combination of genetic and environmental factors [1]. Thus, it is critical to focus on modifiable risk factors for controlling hypertension. The evidence suggests that blood pressure can be lowered by adopting a dietary pattern such as the DASH diet, eating less saturated fat and total fat, obtaining enough potassium, limiting salt intake, and limiting alcohol consumption [4,5].

The DASH eating plans offer dietary recommendations based on calorie requirements, with a focus on the consumption of grains, fruit, vegetables, lean meat, fish, dairy, and nuts and a limited consumption of sweets, sugar-sweetened beverages, and saturated fats [4,6,7]. The eating plan was created to increase fibre, calcium, magnesium, and potassium intake while lowering cholesterol [6]. An umbrella review of systematic reviews and meta-analyses has demonstrated that the DASH diet, either alone or in combination with other lifestyle changes—including sodium restriction, weight loss, or physical activity—is effective for lowering BP levels [8]. Even though reducing BP lowers the risk of kidney and heart disease, most people struggle to control their BP with medication or lifestyle changes [9,10]. Self-management is one of the most effective strategies for dealing with hypertension, as it allows people with hypertension to take more responsibility for their own health [11]. Dietary self-management often requires users to record their meals in order to help control the consumption of calories and specific nutrients, i.e., carbohydrates or cholesterol [12]. Healthcare professionals will often advise patients to keep a diet diary, the key argument for this being that it assists patients in changing their behaviour by encouraging them to take more responsibility for their health and to reflect on their eating habits [12].

The current increase in information and communication technology, such as mobile health, has aided chronic condition self-management [13,14]. Smartphone sales worldwide exceed 2 billion per year, and there are more than three billion smartphone users today [15]. Mobile apps support clinical intervention strategies by leveraging the multifunctional capabilities of mobile devices [16]. Mobile health interventions are promising in low- and middle-income countries regarding promoting physical activity and healthy diets [17]. Several systematic reviews and meta-analyses have assessed the role of dietary monitoring for chronic disease management [18] and changing dietary behaviour [19] using mobile phone applications. They found that dietary monitoring has positive effects on managing chronic diseases, particularly weight loss and changes in dietary behaviour. Also, adherence to dietary self-monitoring via apps may help registered dietitians determine patients’ dietary patterns and recognise factors supporting or impeding goal attainment [20].

Many smartphone apps are already readily available to hypertensive patients, and the number is rapidly growing [21,22]. While most of these apps are designed to assist people with hypertension with management and control [21,22], a systematic review by Alessa et al. [23] found that few studies report the effectiveness of apps in supporting the self-management of hypertension. Similarly, there are only a few studies on the effectiveness of apps to help patients adhere to the DASH diet. For example, a recent systematic review [24] found only five studies that met the inclusion criteria and evaluated DASH diet app interventions to improve DASH diet self-management.

Self-management programmes that are accompanied by theory-based interventions have been demonstrated to be more effective [25,26]. Behaviour change techniques (BCTs) are an important feature of nutrition apps since such techniques involve observable and reproducible components that can be used to alter behaviour [20]. The theoretical frameworks provided by such programmes enable the identification of target behaviours and behavioural modification strategies required to attain desirable health outcomes. However, according to studies, many commercial health apps lack the theoretical underpinning and consistent use of BCTs [23,27].

The shortcomings raise serious concerns about health apps, which may provide little or no benefit and may even pose a risk to users [28], thus underscoring the importance of providing adequate information to patients and healthcare professionals about the effectiveness and quality of these apps, as well as the strength of their privacy and security measures. In addition, the findings of Alessa et al. highlight the value of describing and exploring the probable theoretical mechanisms of action in commercially available apps [23]. The Theoretical Domain Framework (TDF) and BCT Taxonomy v1 (BCTTv1) have been widely used to define BCTs in health interventions [23,29] and to study the probable theoretical mechanisms of action by grouping BCTs with TDF mechanisms of action, particularly those BCTs related to chronic diseases.

However, app quality involves more than effectiveness, and there has been considerable debate about how to define it, with many frameworks available. In 2016, a review and content analysis by DiFilippo et al. assessed the availability and quality of DASH diet applications in Apple’s app store in the US using the App Quality Evaluation (AQEL) [30]. They found a lack of free apps supporting DASH education, and the free apps that were available needed improvement, while paid apps could be beneficial in supporting DASH education. Furthermore, most health applications were found to lack suitable privacy and security safeguards to protect users’ data, thereby posing a threat to user confidentiality [3,23,31], thus jeopardising both users’ personal information and their trust in the app.

We carried out a systematic review to evaluate the effectiveness of smartphone apps that support dietary self-management to improve adherence to DASH diets and, consequently, lower blood pressure [24]. Five studies met the inclusion criteria and were included (three RCTs and two pre-post studies), two of which were conducted on apps that are available on app stores (NOOM and Nutritionix Track). This study found that there is weak emerging evidence that DASH smartphone apps improve adherence to DASH diets and consequently lower blood pressure.

To date, only one study has examined the quality of DASH diet apps in the US Apple App Store [30]. No previous review has provided a comprehensive analysis of DASH diet self-management apps that are available on the most popular platforms: iPhone (Apple App Store) and Android (Google Play) stores, including the quality of apps, their privacy and security, and their behavioural change mechanisms, giving an indication of their potential effectiveness. As such, there is a lack of evidence available to guide the choice of which apps are suitable to support self-management of the DASH diet. Therefore, our research question in this study was: Which generally available apps are suitable for use in DASH diet self-management, based on their likely effectiveness, quality, and security?

## 2. Materials and Methods

### 2.1. App Identification and Selection

In November 2022, systematic searches were conducted in the UK, specifically in the iPhone (Apple App Store) and Android (Google Play) stores. These two platforms were chosen because they are the most widely used operating systems worldwide [32]. Apps were identified using the following keywords: ‘DASH diet’, ‘high blood pressure diet’, and ‘hypertension diet’. These search keywords were entered individually, with no specific search categories such as health and fitness, into the App Store and Google Play database search bars.

For screening, each app’s title, description, and screenshots were considered by the primary researcher (GA) using an iPhone 7 (version iOS 14.4, Apple Inc., Cupertino, CA, USA) to identify apps in the Apple App Store and an Android Samsung Galaxy S20 5G (version Android 12, Samsung Electronics, Suwon, South Korea) to identify apps in the Google Play Store. The app inclusion criteria were as follows: (1) the app allows the user to track their diet intake; (2) the app and the description of the app were written in English; and (3) the DASH diet was included in the app description. The app exclusion criteria were as follows: (1) apps not related to the DASH diet; (2) apps not meant for self-management (i.e., offering only information to promote the DASH diet, DASH diet recipes and plans, etc.); (3) apps offering a variety of diets (i.e., DASH is not the primary focus); (4) apps designed for doctors and dietitians to use in their professional work; and (5) apps for food sales or takeaway orders. Average user ratings were not employed as a selection criterion in the current study, in contrast to earlier studies that assessed the content of health apps by selecting apps based on user reviews [33,34], as their subjectivity may not always provide relevant information on app quality [35].

If an app appeared on both platforms, either version could be used for testing. Apps that appeared many times during the search process were only listed once. All relevant apps were downloaded, including free and paid apps. The researcher selected the paid version of an app if the paid version included increased functionality. The apps were then checked to see whether or not they met the inclusion criteria. Finally, the apps were run for three days if they fulfilled the criteria, so that the researcher could investigate whether the app provided any reminders or notifications.

In addition, commercially available apps found in the previous (updated) systematic literature review were included if they were not found in the app store searches [24].

### 2.2. Data Extraction 

The data extraction process involved using a Microsoft Excel spreadsheet to compile information from the apps. Each of the apps’ name, developer, price, functionality, and recent updates and versions were all noted on the spreadsheet.

### 2.3. In-Depth Analysis

All the apps were downloaded and evaluated by four reviewers—one registered dietitian (EN), one nutrition epidemiologist (specialising in online dietary assessment tools, SA), one nutritionist (GA), and one mHealth specialist (TA). The app’s quality was assessed by three reviewers (EN, SA, and GA), while privacy, security, and likelihood of effectiveness were evaluated by two reviewers (TA and GA).

#### 2.3.1. Likelihood of Effectiveness of the DASH Diet App and Theoretical Underpinnings

As few DASH diet apps have been evaluated for effectiveness [24], the presence of Behaviour change techniques (BCTs) was quantified to indicate the likelihood of effectiveness. The apps were examined to determine whether they were supported by theoretical frameworks, referred to as TDF mechanisms of action. To determine the mechanisms of action that underpin existing apps, the BCT Taxonomy v1 was used to code each app’s content and extract the number of BCTs in each app, as well as their frequency of use [36]. Each BCT was coded as (0) if it was absent and (1) if it was present [23]. The two reviewers (GA and TA) conducted the analyses independently. The Cohen’s kappa for each item was calculated and used to determine the interrater reliability for the presence or absence of BCTs.

The BCTs were then mapped onto the TDF mechanisms of action based on expert consensus that has been reported in previous studies relating BCTs to TDF domains for health interventions, and all researchers in this study agreed with this mapped judgement [29,37]. The two reviewers (GA and TA) worked independently to link the BCTs to the TDF in order to explore the domains underpinning the app. Disagreements were resolved by discussion with other researchers (LdW, EW, MSH).

#### 2.3.2. General App Quality

The quality of each app was evaluated using the AQEL tool. This assessment tool is used to measure the quality of apps that aid in the development and selection of nutrition interventions [38]. AQEL has five domains: Educational domains (behaviour change potential, knowledge acquisition support, and skill development) address how the app will increase knowledge, improve skills, and change behaviour. Meanwhile, the other two domains deal with the app’s functionality, including its speed, colours, icons, and overall purpose, which should have a clear name and description [38]. There are also two modifiable domains: the app’s suitability for the target age group (adults) in terms of nutrition needs and cognitive ability and its relevance to the target audience (in this case, those seeking help with DASH diet self-management to control hypertension). Notably, this tool has been used to evaluate the quality of existing nutrition apps [30,39,40].

The reviewers (SA, EN, and GA) downloaded the selected DASH apps and familiarised themselves with each app and its features before beginning the evaluation. Then, they completed the AQEL survey in Qualtrics (version 2016–2017, Provo, UT, USA) individually. As each domain contained a different number of questions, all sum scores were translated to a 10-point scale for comparison. For each app, the average score for each domain evaluation was calculated. Apps receiving a score of eight or more in most domains were considered high-quality [30].

#### 2.3.3. Privacy and Security

Privacy and security were assessed in accordance with the guidelines set by the Information Commissioner’s Office in the UK and the Online Trust Alliance [41,42]. These guidelines are composed of eight questions that can be answered with yes, no, or not specified/not applicable to evaluate the accessibility and availability of privacy policies, data sharing, collection methods, and data security, as defined by the privacy and security statement. The apps were assessed independently by two researchers (GA and TA), and disagreements were addressed through discussions with other researchers (LdW, EW, MSH).

### 2.4. Statistical Analysis

SPSS software (IBM SPSS Statistics for Windows, Version 26.0. IBM Corp., Armonk, NY, USA). was used for all of the analyses. The frequencies of BCTs and TDFs were generated as mean and standard deviation (SD). To examine interrater reliability, two-way random absolute intraclass correlation coefficients (ICCs) utilising average measures were used to assess the continuous data. Good and excellent agreement were defined as an ICC between 0.75 and 0.90 and greater than 0.90, respectively, while moderate agreement was defined as between 0.50 and 0.75 [43]. Also, Cohen’s kappa was used to assess the interrater reliability of the ordinal data. Perfect agreement was defined as 0.81–1.00, and moderate agreement was defined as 0.61–0.80 [44]. The SD, mean, and median of BCTs and TDFs were calculated for the reviewed apps.

## 3. Results

### 3.1. Identification of DASH Diet Self-Management Apps in the Published Literature and App Stores

A total of 659 apps were found after searching the two app platforms (419 in the Android Google Play Store and 238 in the iPhone Apple App Store). A previous systematic review identified two apps—NOOM and Nutritionix Track—[24] that the app store searches did not capture, as neither of the apps’ titles nor descriptions mentioned the DASH diet. However, both of these apps have been used in interventional studies targeting DASH diet self-management [24]. Therefore, both applications were included. The apps’ titles and descriptions were reviewed for eligibility. A total of 647 apps were excluded because they did not meet the criteria for inclusion. The remaining 12 apps were then analysed further. A further five apps were excluded because their basic functions did not work (n = 1), the information was related to other diets (n = 1), they offered a variety of diets (i.e., DASH was not the primary focus; n = 1), or they were linked to research projects requiring an access code (n = 2). Finally, this review included seven apps: six free (in-app purchases) and one paid. Figure 1 illustrates a summary of the selection of apps using the PRISMA flow diagram.

### 3.2. Characteristics of the Selected Apps

Three out of the seven apps that fulfilled the selection criteria were accessible on both platforms. There were three apps available on iPhone only and one app available on Android only. The data for each app are presented separately and can be found in the Appendix A.

One of the seven apps was a fee-based download, at a cost of £0.81. The remainder of the apps were available for free download, although they required a monthly subscription to access all the features, new meal updates, personal plans, or extra measurements, including BMI, waist measurements, and heart rate. The costs ranged between £1.90 and £121.82 per item or for the programme. One app also required a one-time purchase of a premium tracking service that estimates vitamins and minerals in food. All the apps were in English, while some of them were also available in Spanish, German, and Korean. The apps examined were released between 2011 and 2020.

Furthermore, all the apps allowed for the quantity of food consumed to be recorded and provided feedback on progress. These apps allow users to enter their food intake data using a variety of methods. Three apps provide a comprehensive database with information from the USDA Food Composition Database, chain restaurant statistics, and other sources (Figure 2). Users can add restaurant meals not already included in this database as well as create custom meals or recipes to simplify the entry of home-cooked meals that they consume regularly. Two apps allow users to select meal categories (breakfast, lunch, etc.) without specifying the quantity or DASH diet food groups, e.g., vegetables or fruit. The remaining two apps include tools to track the intake of DASH diet food groups (Figure 2). Some include images to illustrate nutritional information, such as graphs to describe the food’s nutritional value. Some apps also include credits (points) or a facial expression symbol (an emoji) to illustrate when users meet a DASH diet serving recommendation.

### 3.3. The Nutritional and General App Functionalities

Table 1 illustrates the nutritional and general app functionalities.

### 3.4. Behaviour Change Techniques and Theoretical Domain Framework

#### 3.4.1. The Presence of Behaviour Change Techniques

This study identified 19 BCTs in the seven reviewed apps. The coding of BCT presence or absence achieved an ‘almost perfect’ agreement between the reviewers (Cohen’s kappa 0.84).

The maximum number of BCTs in one app was 19, and each app contained at least eight BCTs. The mean number of BCTs was 12.7 (SD 3.6). Eight BCTs were present in all seven apps reviewed: “self-monitoring of behaviour”, “self-monitoring of outcome of behaviour”, “problem solving”, “action planning”, “feedback on outcomes of behaviour”, “review behaviour goal”, “goal setting (behaviour), and “review outcome goal”. One of the nineteen BCTs, categorised as “social incentive”, was only present once. Appendix A provides an overview of the frequency of common BCTs.

#### 3.4.2. Mechanisms of Action of the Theoretical Domain Framework

The BCTs found in the seven apps reviewed could be mapped onto nine different TDF mechanisms of action. Each app had a different number of TDF mechanisms of action, ranging from six to nine, with a mean of 7.7 (SD 1.1). Table 2 presents the frequency of TDF mechanisms of action used in the seven apps. Further information can be found in Appendix A.

### 3.5. General App Quality

Three reviewers (SA, EN, and GA) used the apps before evaluating them. Interrater reliability (n = 3) was deemed to be good (ICC > 0.75) for six apps and excellent for one app (ICC = 0.92). Only three apps (DASH To TEN, NOOM, and My Dash Diet: Food Tracker and low sodium Recipes) received high scores (>8 of 10) across four out of seven AQEL quality domains, while the remaining apps received low scores (<8 of 10) across most of the AQEL quality domains (Table 3). Each of the three high-scoring apps received high scores in the AQEL domains of ‘knowledge building’ and ‘appropriateness for adults’. Regarding app appropriateness for high blood pressure patients, only the My Dash Diet: Food Tracker and low sodium Recipes app received a high score because it was found to track food intake and sodium consumption, blood pressure, body measurements, macro- and micro-nutrients, water intake, weight, and physical activity. Although the other two apps received a lower score on their appropriateness for supporting DASH diet self-management for hypertensive patients, with scores of 7.5 and 7.1, their scores were still higher than those of the other apps. This was because both apps, DASH To TEN and NOOM, tracked food intake, but they did not automatically calculate sodium consumption and instead only provided information about the amount of sodium consumed. However, both tracked physical activity and water intake. NOOM also reported BP readings but provided no specific information regarding the DASH diet, although it adopted a similar method to the DASH diet and its lessons and strategies were well suited to the DASH diet recommendation [45].

### 3.6. Data Privacy and Security

#### 3.6.1. Availability and Accessibility of Privacy Policy

Of the seven apps in this study, six had a privacy policy, and one (DASH Diet Tracker) did not. None of these six had applied a short-form privacy and security notice describing the key data practices detailed in the entire privacy policy, and this may be because the policies had already been written concisely. Multilingual policy was uncommon; only the NOOM app provided the policy in four other languages (Korean, Spanish, and German).

#### 3.6.2. Data Gathering and Sharing

All the apps disclosed the fact that they collected personally identifiable information, such as email or age. For five apps, the developer reported that they share the data they collect with third parties and explained the data-sharing practice. Only one app (My Dash Diet: Food Tracker and low sodium Recipes) disclosed that it does not share user information with anyone.

#### 3.6.3. Data Security

Four apps described how the user’s data were kept secure: NOOM, DASH To TEN, Nutritionix track, and DASH Diet: Doctor Recommendation. The privacy policy stated that data security and privacy were critical to their operations and that user data were encrypted, anonymised, or only viewed by authorised individuals. Only one app (My Dash Diet: Food Tracker and low sodium Recipes) did not discuss user data security. Overall, four apps met data privacy and security standards. Data gathering, sharing, and security as detailed in the privacy policies can be found in the Appendix A.

### 3.7. Selection Process

All seven apps identified as potentially supporting DASH diet self-management are shown in Table 4. The most suitable apps were selected based on their likely effectiveness, adequate privacy and security, and high quality.

All apps have a theoretical basis, and of the nine Theoretical Domain Framework (TDF) mechanisms of action that underlie these apps, between six and nine are involved in each app. Thus, all seven apps were deemed likely to be effective. Regarding app quality, four apps were excluded because they scored low (App Quality Evaluation (AQEL) < 8) in most domains. Three apps were excluded because one had no privacy policy and two did not adequately protect user data. Therefore, NOOM and DASH To TEN were the only apps that met the selection criteria.

## 4. Discussion

### 4.1. Principal Findings

Dietary apps have the potential to improve dietary behaviour. However, these apps must be effective, secure, and high-quality to achieve their potential. The number of DASH diet apps that provide dietary self-management is limited. Only seven apps were found that met the inclusion criteria, including supplying dietary self-monitoring and feedback that contributes to enhancing diet outcomes [19]. The results of this study demonstrate the risks of commercial app availability. Although all seven apps are built using established theory, only two apps (NOOM and DASH To TEN) were deemed high-quality and demonstrate sufficient privacy and security measures. Only one of these apps (NOOM) has been assessed for efficacy in a trial [46]. Addressing gaps in app development (such as safety concerns and a lack of quality) is essential for improving app quality, privacy, security, and effectiveness. One way of achieving this is by increasing engagement with researchers and experts [47,48]. These improvements would increase both consumer trust and provider confidence [47,48].

### 4.2. App Functionalities

All apps in this review provided interactive features, including dietary self-monitoring and feedback, and some offered educational information about the DASH diet and goal setting. These findings agree with another study that reported that self-monitoring, goal setting, and providing feedback were the most frequently identified types of BCTs on weight management apps [49] because these functions are essential components of behaviour change [50]. According to a previous content analysis of weight management apps with interactive features, apps that provide features such as tracking behaviours were associated with higher engagement scores across the Mobile App Rating Scale domain [49]. In addition, some of the apps also sent notifications to remind users to log their food intake to increase engagement with the behaviour change app [51]. Interactive functions in dietary apps may improve engagement and effectiveness [49,50]. Therefore, interactive functionality should be included in the development of future apps to promote DASH diet self-management.

The apps included various approaches that users could take to log their food intake. For example, in some apps, users could calculate their food servings based on DASH recommendations; however, the portion sizes were described textually without images or icons that could help the user determine the correct serving size, as was also found by a previous study that assessed popular nutrition apps [52]. Qualitative research exploring the barriers to following the DASH diet among Black Americans has shown that most patients faced difficulty calculating their food servings because it was unfamiliar [53]. Therefore, dietary apps should provide a serving size guide to assist users in calculating their food serving size.

In other apps, users could select meal categories (breakfast, lunch, etc.) without specifying the quantity they consumed or the DASH diet food groups (vegetables, fruit, etc.) to which the foods belonged. Although being able to choose the meal category simplified the selection process, this approach is imprecise since some users prefer a more detailed approach [12].

Alternatively, users could select food via text search from large food databases or barcode scanner technologies, which is the most common method of recording food intake among nutrition apps [12]. König et al. reported that participants expressed satisfaction with using a comprehensive food database to self-monitor their food intake [54]. Further qualitative research is needed to explore the best approaches to food logging.

### 4.3. Likelihood of Effectiveness and Theoretical Underpinnings of the DASH Diet Apps

The effectiveness of only two of the reviewed apps was supported by published studies [24], and only one of these apps provided evidence about its effectiveness and discussed and demonstrated the necessity of such evidence in its app store description. In the absence of direct evidence of effectiveness for most apps, we used the presence of BCTs to indicate the likelihood of effectiveness. All apps utilised BCTs, which mapped onto TDF mechanisms. Nineteen BCTs overall, ranging from eight to nineteen per app, were identified. The current study’s finding in this regard is higher than the average number of BCTs found in prior app reviews, which ranged from two to eight in apps aimed at changing adults’ physical activity and dietary behaviours [55,56]. Moreover, a systematic review of studies investigating the effect of apps on nutrition behaviours and nutrition-related health outcomes found that apps included a range of 2–11 BCTs [57].

In the current review, the most common BCTs were feedback, monitoring, goals, and building knowledge. These self-regulation strategies have been demonstrated to be effective in enhancing dietary behaviour [18,19,58]. This study’s finding agrees with the findings of a systematic review and meta-analysis that evaluated the effectiveness of nutrition-related app-based mobile interventions, highlighting that the majority of interventions involved four clusters of Behaviour change techniques (BCTs), namely “goals/planning”, “feedback/monitoring”, “shaping knowledge”, and “social support” [57]. Furthermore, previous work suggests that dietary behaviour change apps targeting weight management need to include goal setting, monitoring behaviour, and providing feedback on that behaviour since these BCTs encourage app users to improve their dietary behaviour [59].

The impact of the number of BCTs on intervention efficacy is still inconclusive; some studies indicated that interventions that include more BCTs seem to have a greater impact than interventions with fewer techniques [60], while others observed no effect [61]. Moreover, certain BCTs may be more effective when used in combination with others [62]. The employment of a variety of BCT groups, as well as the techniques used within each BCT group, could theoretically improve effectiveness by tackling dietary self-management barriers.

The use of a theoretical framework is critical to developing behaviour change interventions [63]. The assessment of the theoretical underpinning of the apps revealed that the 19 BCTs identified in the seven reviewed apps linked to nine TDF mechanisms of action that may support DASH diet self-management. Our results, supported by previous research, identified the main TDF domains that could support chronic disease self-management and influence patients’ behaviours [23,64,65]. Which combinations of BCTs or TDF mechanisms improve chronic disease self-management, however, is unknown [66,67]. There have been limited investigations into how existing apps map BCTs to TDF domains. The current study’s linking of BCTs to TDF mechanisms of action may therefore assist developers and researchers in identifying suitable BCTs when developing apps and determining which BCTs are most effective and why [68,69].

### 4.4. General App Quality

The quality assessment of the apps that supported DASH diet self-management indicated the need to improve their quality. Only three apps scored well in the majority of the AQEL domains. However, none of the included apps scored well (AQEL > 8) in all educational domains (behaviour change potential, knowledge building, and skill building). This finding is consistent with a previous study that evaluated apps that supported DASH diet education and found that few apps were of sufficient quality in this area [30]. It is essential to educate patients about self-management to support chronic disease management, and they need ongoing help to improve their self-management skills and achieve the desired behavioural change [70]. Therefore, it is imperative to improve the educational domains of commercial applications of the DASH diet.

Regarding appropriateness for hypertensive patients, the current study found that most DASH diet apps scored low and that only My DASH Diet scored well (AQEL > 8); this is because it tracks sodium intake, a critical factor in reducing blood pressure [71].

### 4.5. Data Privacy and Security

Due to the large size of the app market, regulatory control of data protection is difficult [23,27]. As a result, app developers are responsible for guaranteeing data privacy and security [72]. The examination of the apps’ privacy policies revealed that the privacy and security of users’ data could be significantly improved, which is consistent with the results of a previous study [23] that assessed apps supporting the self-management of hypertension and found that many lacked privacy policies, most of which violated users’ privacy and data security. The key advantage of collecting and analysing users’ data via app is that developers may be able to improve their products using the data [73]. However, information regarding these practices should be accessible and easy to understand so that potential users can decide whether to download and install the app [74].

### 4.6. Strengths and Limitations

The systematic approach used for app identification and assessment is one of this study’s strengths. Six hundred and fifty-nine apps, both free and fee-based, were screened across the two most common mobile phone platforms to identify a sample. The sample was assessed by four independent reviewers with diet, nutrition, and mHealth expertise. The evaluation also covered various aspects, including quality by using validated AQEL, privacy and security, and likely effectiveness. Additionally, this is the first systematic review to investigate the apps’ theoretical underpinnings by mapping BCTs to TDF mechanisms. For transparency purposes, a list of the apps analysed has been provided. Providing an app list has been recommended as a good method to conduct quality health-related app reviews, as it might assist users and healthcare providers in making informed choices as well as inspire app developers to improve their app’s content [75].

This study has several limitations. Firstly, app stores are not designed to conduct robust, rigorous searches in the same manner as electronic journals. Furthermore, due to the nature of search algorithms and personalised app content in commercial app stores, it may be difficult to remove duplicates directly from app store searches [76]. Therefore, it can be challenging to repeat the search strategy [76]. Currently, there are no established guidelines for conducting and reporting systematic searches of app stores. However, efforts are being made to reach an agreement on such guidelines [76]. Secondly, our search was limited to publicly accessible apps, and we did not contact any developers whose apps required a code to access. Thirdly, this review only focuses on the two most common platforms in the UK marketplace. Other countries could offer DASH diet apps with different features, qualities, and BCTs. Fourthly, this review did not consider apps available in languages other than English. Therefore, it is necessary to conduct further research on other marketplaces and languages. Fifthly, we evaluated apps for eligibility by checking their titles and descriptions against the inclusion and exclusion criteria. As a result, we may have missed apps that did not include our search terms in their titles and descriptions. To some extent, these issues were overcome by including commercial apps found in a recent systematic review that assessed the effectiveness of DASH diet apps that support dietary self-management. We also updated the search to ensure no recently published studies were missed. Sixthly, if the quality and security assessment had been conducted by app users instead of researchers or by individuals with characteristics different from the assessors who conducted the current evaluation, the results might have been different. To overcome these issues, four researchers independently evaluated the apps, resulting in high interrater reliability and thereby limiting bias. Moreover, apps were assessed within three days, and some app features and BCTs might have been overlooked, such as follow-up prompts, which can require prolonged usage. Therefore, future studies should examine the optimal time frame for assessing an app’s quality and whether BCTs are present. Finally, the evaluation of data privacy and security was limited to the analysis of the apps’ policy statements, despite the fact that there is evidence of a misalignment between policy statements and app developers’ actual practices [31].

### 4.7. Implications

More studies are needed to evaluate DASH diet apps. It is currently unclear which app features or characteristics effectively increase users’ diet adherence. Therefore, RCTs are needed to determine the effectiveness of theory-based apps. Additionally, qualitative research is necessary to examine users’ experiences with apps and BCTs. Furthermore, the optimal number of BCTs and TDF mechanisms of action for dietary apps is unknown. Thus, more studies are needed to investigate the effectiveness and suitable number of BCTs and TDF mechanisms of action to increase dietary adherence. Little information is available about how users incorporate these apps into their daily routines or the facilitators and barriers to increasing diet adherence using apps. Future studies are needed to assess the acceptability and usability of the highest-quality apps that support DASH diet self-management.

The findings of our study also have practical implications for multiple stakeholders. First, dietitians, health care professionals, and potential users should be aware of the limitations regarding the security of personal data and the quality of DASH diet apps in supporting DASH diet self-management. Second, using our findings, dietitians may have sufficient information to suggest a high-quality app to patients to support DASH diet self-management. Finally, the findings can help smartphone application developers address privacy, security, and app quality to improve the current market.

## 5. Conclusions

Although DASH diet apps are widely available, few apps support the self-management of the DASH diet. All seven apps with self-monitoring functionality had a sufficient theoretical basis to demonstrate their likely effectiveness. However, most had significant shortcomings in their quality and data security. Only NOOM and DASH To TEN were distinguished as high-quality because they scored well in most AQEL domains and demonstrated sufficient privacy and security measures. However, none scored well (AQEL > 8) in all the educational domains essential for improving dietary self-management skills. Future research is needed to evaluate the acceptability, usability, and effectiveness of these high-quality apps. Additionally, more efforts are needed to improve the quality of DASH diet apps, especially in the education domain, with data security and interactive functions to help high blood pressure patients manage their diet.

## Figures and Tables

**Figure 1 nutrients-15-03476-f001:**
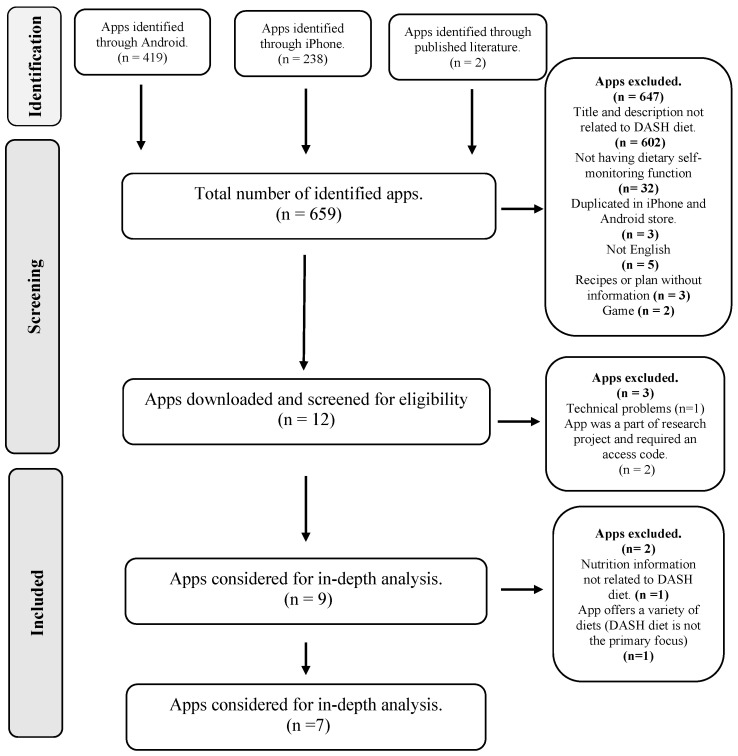
Flow diagram of app search and selection.

**Figure 2 nutrients-15-03476-f002:**
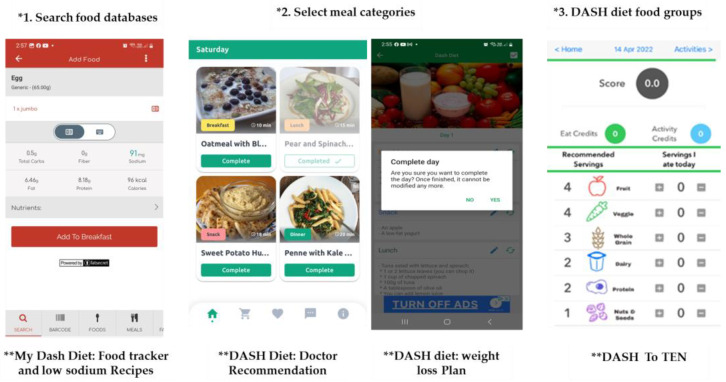
The food tracker Methods; * the food tracker methods; ** the app’s name.

**Table 1 nutrients-15-03476-t001:** The nutritional and general app functionalities.

Name of App	App Functions
My DASH Diet: Food Tracker and low sodium Recipes	Self-monitoring (food, water, BP, physical activity, and weight). Communication with friends by email. Feedback (macronutrients, micronutrients such as sodium intake, weight, BP, and water). Educational information uses credible sources. Goal setting (sodium intake and weight). Recipes and shopping list. Intelligent recognition system to identify food barcodes and search functionality.
NOOM	Self-monitoring (food, water, BP, blood glucose, physical activity, weight). Communication with coach through the in-app chat. Feedback (calorie density, weight, and BP). Educational information uses credible sources. Goal setting (weight). Notifications to log food and read the articles and push notifications to indicate the user has high blood pressure and needs to see a doctor. Quiz to assess users’ information. Intelligent recognition system to identify food barcodes and search functionality.
Nutritionix Track	Self-monitoring (food, water, BP, physical activity, and weight). Communication with coach through the in-app chat. Feedback (macronutrients, micronutrients such as sodium intake, weight, and water). Goal setting (weight). Notification to log food. Intelligent recognition system to identify food barcodes and search functionality. Uses pictures of food.
DASH Diet Tracker	Self-monitoring (food and weight). Feedback (food group intake). Goal setting (weight).
DASH diet: weight loss Plan	Self-monitoring (to follow DASH diet plan and weight). Educational information: no provided sources. Feedback (follow DASH plan and weight). DASH diet plan. Shopping list. Notification (to track DASH plan and weight).
DASH Diet: Doctor Recommendation	Self-monitoring (to follow DASH diet plan and weight). Communication with app team through the in-app chat. Feedback (follow DASH plan and weight). Educational information uses credible sources. Goal setting (weight). Recipes and shopping list. Notification of dietary plan.
DASH To TEN	Self-monitoring (food, water, sleep, medication, meditation). Feedback (food group intake and weight). Goal (weight). Educational information uses credible sources. DASH diet plan. Notification to log food and push notifications to indicate the user has exceeded the DASH diet recommendations.

**Table 2 nutrients-15-03476-t002:** The Theoretical Domain Framework mechanisms of action mapped Behaviour change techniques.

The Theoretical Domain Framework	Behaviour Change Techniques	Frequency of TDF Domain in Apps (N)
Beliefs about capabilities	Social support	7
	Problem solving	
	Action planning	
Goals	Goal setting (outcome)	7
	Goal setting (behaviour)	
	Review outcome goal(s)	
	Review behaviour goal(s)	
	Action planning	
Knowledge	Information about health consequences	7
	Instruction on how to perform a behaviour	
	Credible source	
	Feedback on behaviour	
	Feedback on outcomes of behaviour	
	Biofeedback	
Skills	Problem solving	7
	Biofeedback	
Beliefs about consequences	Feedback on behaviour	7
	Feedback on outcomes of behaviour	
Behaviour regulation	Self-monitoring of behaviour	7
	Self-monitoring of outcome(s) of behaviour	
	Problem solving	
Memory, attention, and decision processes	Prompts/cues	5
	Habit formation	
Reinforcement	Credible source	4
Emotion	Reduce negative emotion	1

**Table 3 nutrients-15-03476-t003:** Mean App Quality Evaluation to support DASH diet self-management out of 10 (n = 3 Raters).

App Name	App Quality Evaluation Domain						
	Behaviour Change Potential	Knowledge Building	Skill Building	Function	App Purpose	Appropriate for Adults	Appropriate for Hypertension
My Dash Diet: Food tracker and low sodium Recipes	7 (0.2)	8.2 (0)	7.4 (0.6)	7.2 (0.4)	10 (0)	9.3 (0.5)	8.5 (0.4)
DASH Diet: Doctor Recommendation	6.5 (0.9)	4.4 (1.0)	6.7 (0)	7.6 (0.2)	6.1 (0.9)	8.6 (0.5)	3.8 (0)
DASH To TEN	7 (0.2)	8 (0.2)	7 (0.6)	8 (0.6)	8.3 (0.1)	8.6 (0.5)	7.5 (0)
DASH Diet Tracker	2.8 (0.6)	2.5 (0.2)	4.4 (1.9)	4.8 (0.2)	4.4 (0.9)	4.3 (0.5)	0 (0)
DASH diet: weight loss Plan	0.6 (0.2)	0.9 (0)	2.9 (2.5)	4.7 (0.9)	2.7 (0.9)	3.6 (0.5)	0 (0)
NOOM	8.4 (1.2)	9.2 (0.5)	7.8 (0)	9.1 (1.1)	3.4 (0.2)	9.6 (0.5)	7.1 (1.4)
Nutritionix track	5.03 (1.02)	3.2 (0.42)	5.5 (1.5)	7.1 (0.24)	8.3 (0)	8 (0.47)	3.8 (0.24)

Note: Data are presented as the mean (SD).

**Table 4 nutrients-15-03476-t004:** The Apps’ quality, privacy and security, and theoretical underpinning (n = 7).

App Name	Version Type	No of BCTs	TDF Mechanisms of Actions, n	Quality of App, n	Privacy and Security
My Dash Diet: Food tracker and low sodium Recipes	iPhone and Android	14	7	4 domains > 8	X
DASH diet: weight loss Plan	Android	8	8	0 domains > 8	X
DASH To TEN	iPhone	13	9	4 domains > 8	✓
DASH Diet Tracker	iPhone	9	6	0 domains > 8	X
DASH Diet: Doctor Recommendation	iPhone	14	8	1 domain > 8	✓
NOOM	iPhone and Android	19	9	4 domains > 8	✓
Nutritionix track	iPhone and Android	12	7	2 domains > 8	✓

## Data Availability

The data presented in this study are available in the Appendix A.

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
