# Peer review of "Identification of the Most Suitable Mobile Apps to Support Dietary Approaches to Stop Hypertension (DASH) Diet Self-Management: Systematic Search of App Stores and Content Analysis"

_nutrients, 2023, doi:10.3390/nu15153476_

Round 1
Reviewer 1 Report
Re: Identification of the Most Suitable Mobile Apps to Support Dietary Approaches to Stop Hypertension (DASH) Diet Self-management Through a Systematic Search of Two App Stores
The present manuscript reviews DASH diet apps for self-management and is the first systematic review to investigate the apps’ theoretical underpinnings by mapping BCTs to the TDF mechanisms. This review fills an important need to provide evidence on the quality and potential effectiveness of health apps to support translation of dietary guidelines and support sustained behaviour change for improved health. It was overall well written. There are a few comments listed below which highlight areas in which clarity would assist in improving the readability and understanding of the results and conclusions.
1. Although the final line of the abstract reads that only 2 apps were deemed to be of adequate quality and security, the general conclusion appears to convey that there are limited apps to support DASH diet self-management. However, if one app is rated ideally, it could be all that is required to support DASH diet self-management for hypertension. From the conclusions, it is not clear if there is one such app among those reviewed. It would be useful to present the names of the 2 apps that were found to be the best among those available as well as to indicate what aspects based on the results of this review could make them better (perhaps educational domains of AQEL as indicated in the discussion). In the opening of the discussion, it reads that apps must be effective, secure, and high-quality to achieve the potential to improve dietary behaviour. Based on the results, it seems these 2 apps may meet the authors’ criteria for an app to have the potential to improve dietary behaviour. Importantly, it is noted in the manuscript that only few showed evidence of effectiveness (publications) and it is of interest to know if these 2 apps did. Making a conclusion on the 2 best apps reviewed is suggested, which would tie in well with the listed strength of assisting healthcare providers make informed choices.
2. Since for each section of results, the data is not always presented by app title, it is unclear which apps were rated best in each area explored (e.g. Data security is only presented as a summary of number of apps and not by title). Although they are summarized in table 4, it would be helpful to present each set of results for each app so the reader can follow along with the results for each app reviewed. Since My DASH Diet: Food Tracker was listed first in the first set of results presented and was found to contain among the larger number of BCTs, I was following along the results with the impression it was among the best until the final section of the results. Consistently highlighting the results individually for the set of apps reviewed across the areas assessed is suggested to improve readability.
3. It is a bit uncertain as to why NOOM was included as it does not appear to be specific to DASH, as is particularly noted in lines 371-373. Could you please clarify?
Minor Comments:
1. In Introduction when referencing the description of the foods within the DASH diet, I suggest including the original DASH paper by the DASH Collaborative group: Appel et al. N Engl J Med. 1997 Apr 17;336(16):1117-24. doi: 10.1056/NEJM199704173361601.
2. The app names in Figure 2 do not match those in Table S1.
3. Unclear if Nutritionnix has DASH-related information. In Table S1, there is no information on DASH. It appears as only a self-calorie tracker. Could you please comment on how it integrates DASH information/tracking?
4. Line 327, should this read seven apps as opposed to five?
5. In lines 178-180 it reads all apps were evaluated by 4 reviewers, however in line 211, only 3 initials are included in the list of reviewers who downloaded the selected DASH apps. Could you please clarify how the 4th reviewer was part of the in-depth analysis if they did not download the apps?
6. For the 2 apps linked to research projects, were you able to reach out to the researchers to inquire as to whether their app would potentially be released post-study? It would be particularly useful to know if an evidence-based app is upcoming and whether the researchers would be willing to have their app included in the present review.
7. For the 2 apps excluded because they had nutrition information related to other diets, could you please confirm if this did not include the DASH diet? Since you pass them upon screening for eligibility, it is not clear if they may have included the DASH diet as well as others. Further, your exclusion criteria in an earlier exclusion box in Figure 1 described an exclusion as “not related to the DASH diet” so it is not clear why in this last exclusion box it would be worded differently.
8. In line 324, it reads that BCTs were mapped to 14 TDF mechanisms of action and refers to Table 2, however there are only 9 TDFs in Table 2 and the abstract reads the 19 BCTs were linked to 9 TDFs. Please verify if the 14 should be 9.
9. In the Supplement: Table S1:
o 9a) unclear which ones were included in the 7 reviewed (abstract reads 1 paid and 6 free with in-app purchases)
o 9b) I believe the developer of My DASH Diet: Food Tracker and Low Sodium recipes is “Prestige” not “Presting”
Author Response
1 |
Although the final line of the abstract reads that only 2 apps were deemed to be of adequate quality and security, the general conclusion appears to convey that there are limited apps to support DASH diet self-management. However, if one app is rated ideally, it could be all that is required to support DASH diet self-management for hypertension. From the conclusions, it is not clear if there is one such app among those reviewed. It would be useful to present the names of the 2 apps that were found to be the best among those available as well as to indicate what aspects based on the results of this review could make them better (perhaps educational domains of AQEL as indicated in the discussion). In the opening of the discussion, it reads that apps must be effective, secure, and high-quality to achieve the potential to improve dietary behaviour. Based on the results, it seems these 2 apps may meet the authors’ criteria for an app to have the potential to improve dietary behaviour. Importantly, it is noted in the manuscript that only few showed evidence of effectiveness (publications) and it is of interest to know if these 2 apps did. Making a conclusion on the 2 best apps reviewed is suggested, which would tie in well with the listed strength of assisting healthcare providers make informed choices.
|
Thank you for these insightful comments, which will help to improve our manuscript. These comments were addressed as follows. In the abstract 1) The names of the two apps were presented. Please see page 1, lines 40-44. In the Discussion 1) Both apps’ names were written, and a peer-reviewed study assessing the effectiveness of the NOOM app was included. See first paragraph, page 16, Lines 461-469. In the Conclusion 1) The names of the two apps were presented, followed by an explanation of how these apps could be better. Please see page 20, Lines 639-649.
|
2 |
Since for each section of results, the data is not always presented by app title, it is unclear which apps were rated best in each area explored (e.g. Data security is only presented as a summary of number of apps and not by title). Although they are summarized in table 4, it would be helpful to present each set of results for each app so the reader can follow along with the results for each app reviewed. Since My DASH Diet: Food Tracker was listed first in the first set of results presented and was found to contain among the larger number of BCTs, I was following along the results with the impression it was among the best until the final section of the results. Consistently highlighting the results individually for the set of apps reviewed across the areas assessed is suggested to improve readability.
|
Throughout the result section, we listed each app's name under each criterion so readers can assess its strengths and weaknesses. |
3 |
It is a bit uncertain as to why NOOM was included as it does not appear to be specific to DASH, as is particularly noted in lines 371-373. Could you please clarify?
|
There are several reasons why the NOOM and Nutritionix Track apps have been added: 1. A systematic review was conducted to determine the effectiveness of smartphone apps in promoting DASH diet self-management. The review discovered two studies utilized commercial apps, namely NOOM and Nutritionix Track, to enhance adherence to the DASH diet and lower blood pressure. 2. Both apps have a self-monitoring feature that enables users to record their meals, which is critical in dietary self-management as it helps to reflect on eating habits. 3. This review emphasizes that the researchers can use self-calorie tracker apps with comprehensive food databases to assist users in recording their food intake. However, combining these apps with other approaches is essential for effectiveness and improving their quality, but these apps need to assess their likely effectiveness and security. We have addressed this issue by adding information explaining why NOOM and Nutritionix Track were added. Please see the result section on pages 6-7, section 3, lines 249-254 |
4 |
Unclear if Nutritionnix has DASH-related information. In Table S1, there is no information on DASH. It appears as only a self-calorie tracker. Could you please comment on how it integrates DASH information/tracking?
|
|
5 |
In Introduction when referencing the description of the foods within the DASH diet, I suggest including the original DASH paper by the DASH Collaborative group: Appel et al. N Engl J Med. 1997 Apr 17;336(16):1117-24. doi: 10.1056/NEJM199704173361601.
|
We have addressed this issue by citing this paper. Please see page 2, line 59. |
|
Comments |
Response |
6 |
The app names in Figure 2 do not match those in Table S1.
|
We have addressed this by writing the complete names of the applications. Please see page 8, Figure 2. The food tracker Methods
|
7 |
Line 327, should this read seven apps as opposed to five?
|
We addressed this by adding the correct number of apps. “Table 2 presents the frequency of TDF mechanisms of action used in the seven apps”. Please see page 11, line 346. |
8 |
In lines 178-180 it reads all apps were evaluated by 4 reviewers, however in line 211, only 3 initials are included in the list of reviewers who downloaded the selected DASH apps. Could you please clarify how the 4th reviewer was part of the in-depth analysis if they did not download the apps?
|
We addressed this by adding what the reviewers did. The app quality was assessed by three reviewers (EN, SA, and GA), while privacy, security, and likelihood of effective use were evaluated by two reviewers (TA and GA). Please see page 5, lines 182-184. |
9 |
For the 2 apps linked to research projects, were you able to reach out to the researchers to inquire as to whether their app would potentially be released post-study? It would be particularly useful to know if an evidence-based app is upcoming and whether the researchers would be willing to have their app included in the present review.
|
The app developers were not contacted. In this study, we examined apps available to the public. We addressed this by adding that in the limitation, “Our search was limited to publicly accessible apps, and we did not contact any developers whose apps required code to access.” Page 19, Lines 595- 597. |
10 |
For the 2 apps excluded because they had nutrition information related to other diets, could you please confirm if this did not include the DASH diet? Since you pass them upon screening for eligibility, it is not clear if they may have included the DASH diet as well as others. Further, your exclusion criteria in an earlier exclusion box in Figure 1 described an exclusion as “not related to the DASH diet” so it is not clear why in this last exclusion box it would be worded differently. |
The DASH diet plan and food tracker app includes general nutrition information that does not focus on the DASH diet. However, after receiving your comment, we found that the education feature did not work when we tried to recheck it. Please see the figure at the end of this document. Regarding the DASH diet recipes app, recently, this app was updated (on 13 March 2023) and added DASH diet information. We screened the App Store in November 2022. However, this app would still not be included because it offers a variety of diets, and users can edit their plans. One of our exclusion criteria was apps offering a variety of diets. Please see the figure at the end of this document (see page 6 of this document). We have amended Figure 1: the flow diagram and the supplementary materials in Table 1.
|
11 |
In line 324, it reads that BCTs were mapped to 14 TDF mechanisms of action and refers to Table 2, however there are only 9 TDFs in Table 2 and the abstract reads the 19 BCTs were linked to 9 TDFs. Please verify if the 14 should be 9.
|
We have addressed that by correcting the number. Please see page 11, line 344. “The BCTs found in the seven apps reviewed could be mapped to 9 different TDF mechanisms of action.”
|
12 |
In the Supplement: Table S1: o 9a) unclear which ones were included in the 7 reviewed (abstract reads 1 paid and 6 free with in-app purchases) o 9b) I believe the developer of My DASH Diet: Food Tracker and Low Sodium recipes is “Prestige” not “Presting”
|
We addressed this by highlighting the seven reviewed apps and correcting the spelling error. |
Reviewer 2 Report
Thank you for giving me the time to review your manuscript. This manuscript is interesting for Mobile Apps to Support Dietary Approaches to Stop Hypertension Diet Self-management. Regarding the contents, the following revision should be considered.
Title
-Title should include a specific research design.
The abstract
-The abstract should include specific research design descriptions. In addition, app’s quality should be defined in the abstract.
Introduction
Generally, there is no paragraph writing. The background contains many paragraphs. The author should focus on theory building, the problems, and research question paragraphs. The first and second paragraphs should include general information on hypertension and nutritional control for hypertension. Moreover, the third and fourth paragraphs should introduce the research question as the theoretical and conceptual framework, including DASH effectiveness, using applications in international contexts and research questions.
-This research lacks the evidence gap and research questions.
-This study focuses on one hospital. However, there is a lack of reasons why this research focuses on the hospital.
-The introduction should include this study's international contexts and research questions.
Method
-Did this study focus on applications only in English?
-Each criterion for quality assessment included in this study should be explicitly explained.
-Sample calculation should be described clearly.
Discussion
With the same background, the authors should use paragraph writing for logical theory building.
This study should describe the limitation of sampling bias, the results' applicability to other settings, and the future investigation in the limitation part.
In the conclusion or discussion, the study’s strengths should be focused on international readers.
Conclusion
The conclusion section describes the result again. The authors should add a description of news regarding this research field.
Author Response
Comments |
Response |
|
Reviewer 2 |
||
Title |
||
|
Title should include a specific research design. |
We have addressed this by adding the research design. Identification of the Most Suitable Mobile Apps to Support Dietary Approaches to Stop Hypertension (DASH) Diet Self-management: Systematic Search of App Stores and Content Analysis. |
The abstract
|
||
2 |
The abstract should include specific research design descriptions. In addition, app’s quality should be defined in the abstract. |
We have addressed this by adding the research design, and the quality of the apps was defined. Please see the abstract.
|
Introduction |
||
1 |
Generally, there is no paragraph writing. The background contains many paragraphs. The author should focus on theory building, the problems, and research question paragraphs. The first and second paragraphs should include general information on hypertension and nutritional control for hypertension. Moreover, the third and fourth paragraphs should introduce the research question as the theoretical and conceptual framework, including DASH effectiveness, using applications in international contexts and research questions.
|
According to our understanding, the reviewer suggests that the introduction needs to be rewritten in a logical flow. We have carefully reviewed the introduction based on your comment and we feel that, for the most part, the logical flow of its presentation and argument is appropriate to introduce the study we have carried out. Initially, the introduction defined hypertension and the DASH diet, followed by background information on the role of a smartphone app in dietary self-management. We then discussed prior research and established the significance of the research problem before presenting our research question. We have revised the research gap and questions to address this concern to provide greater clarity. Please see page 4, lines 134-138. |
2 |
This research lacks the evidence gap and research questions.
|
This comment was addressed by clarifying the research gap and formulating the research question. “As such, there is a lack of evidence available to guide the choice of which apps are suitable to support self-management of the DASH diet. Therefore, our research question in this study was: Which generally available apps are suitable for use in DASH diet self-management, based on their likely effectiveness, quality, and security? Please see Page 4, lines 134–138 |
3 |
This study focuses on one hospital. However, there is a lack of reasons why this research focuses on the hospital.
|
We assume this comment is based on a misunderstanding, as the manuscript does not mention hospitals. |
4 |
The introduction should include this study's international contexts and research questions. |
According to our understanding, the reviewer wants more focus on the international context. The introduction of this study provided a global perspective on hypertension, highlighting its significant impact on cardiovascular and renal functions. We addressed this comment by adding the potential benefit of mHealth in low-and middle-income countries. “Mobile health interventions are promising in low- and middle-income countries regarding promoting physical activity and healthy diets” Please see page 2, lines 77-79. Also, we added the research question. Please see page 4, lines 134-138. |
Method
|
||
1 |
Did this study focus on applications only in English?
|
We have addressed that by modifying the inclusion criteria” The app and the description of the app were written in English” Please see the methodology section on page 4, line 154. |
2 |
Each criterion for quality assessment included in this study should be explicitly explained.
|
According to our understanding, we addressed that by explaining AQEL domains. Please see pages 5-6, lines 208-216. |
3 |
Sample calculation should be described clearly.
|
According to our understanding, we addressed that by explaining why certain apps were excluded. Please see the figure1 flow diagram. Page 7. |
Discussion
|
||
1 |
With the same background, the authors should use paragraph writing for logical theory building.
|
We have carefully reviewed the discussion based on your comment and we feel that the logical flow of its presentation and argument is appropriate to discuss the study we have carried out. |
2 |
This study should describe the limitation of sampling bias, the results' applicability to other settings, and the future investigation in the limitation part.
|
We have addressed that by describing the limitation of sampling bias, applicability to other settings, and future investigation into the limitation. Thirdly, this review only focuses on the two most common platforms in the UK marketplace. Other countries, such as India, could offer DASH diet apps with different features, qualities, and BCTs. Thirdly, this review did not consider apps available in languages other than English. Therefore, it is necessary to conduct further research on other marketplaces and languages. Please see page 19, lines 597-601. Also, in the methodology, we have justified why we focused on the two platforms in the UK. Please see page 4, lines 143-144. |
3 |
In the conclusion or discussion, the study’s strengths should be focused on international readers.
|
We understand that there may have been a misunderstanding regarding the comment. We just wanted to clarify that the manuscript did not focus on any particular country or nationality. Instead, it was focused on an international context. |
Conclusion
|
||
1 |
The conclusion section describes the result again. The authors should add a description of news regarding this research field.
|
According to our understanding, what do we need in the next step? We have addressed that by adding that “Additionally, more efforts are needed to improve the quality of DASH diet apps, especially in the education domain, with data security and interactive functions to help high-blood pressure patients manage their diet” Please see page 20, lines 647-649 |
Round 2
Reviewer 2 Report
This article was revised to reach the quality of our journal.